# Automated Quantification of Simple and Complex Aortic Flow Using 2D Phase Contrast MRI

**DOI:** 10.3390/medicina60101618

**Published:** 2024-10-03

**Authors:** Rui Li, Hosamadin S. Assadi, Xiaodan Zhao, Gareth Matthews, Zia Mehmood, Ciaran Grafton-Clarke, Vaishali Limbachia, Rimma Hall, Bahman Kasmai, Marina Hughes, Kurian Thampi, David Hewson, Marianna Stamatelatou, Peter P. Swoboda, Andrew J. Swift, Samer Alabed, Sunil Nair, Hilmar Spohr, John Curtin, Yashoda Gurung-Koney, Rob J. van der Geest, Vassilios S. Vassiliou, Liang Zhong, Pankaj Garg

**Affiliations:** 1Norwich Medical School, University of East Anglia, Norfolk NR4 7TJ, UK; r.li2@uea.ac.uk (R.L.); h.assadi@uea.ac.uk (H.S.A.); gareth.matthews@uea.ac.uk (G.M.); b.kasmai@uea.ac.uk (B.K.); marina.hughes2@nhs.net (M.H.); v.vassiliou@uea.ac.uk (V.S.V.); 2Norfolk and Norwich University Hospitals NHS Foundation Trust, Norfolk NR4 7UY, UK; zia.mehmood@nnuh.nhs.uk (Z.M.); c.grafton-clarke@uea.ac.uk (C.G.-C.); vaishali.limbachia@nnuh.nhs.uk (V.L.); rimma.hall@nnuh.nhs.uk (R.H.); kurian.thampi@nnuh.nhs.uk (K.T.); david.hewson@nnuh.nhs.uk (D.H.); marianna.stamatelatou@nnuh.nhs.uk (M.S.); sunil.nair@nnuh.nhs.uk (S.N.); hilmar.spohr@nnuh.nhs.uk (H.S.); john.curtin@nnuh.nhs.uk (J.C.); yashoda.gurung-koney@nnuh.nhs.uk (Y.G.-K.); 3National Heart Research Institute Singapore, National Heart Centre Singapore, 5 Hospital Drive, Singapore 169609, Singapore; zhao.xiaodan@nhcs.com.sg (X.Z.); zhong.liang@duke-nus.edu.sg (L.Z.); 4Division of Biomedical Imaging, Leeds Institute of Cardiovascular and Metabolic Medicine, University of Leeds, Leeds LS2 9JT, UK; p.swoboda@leeds.ac.uk; 5Department of Infection, Immunity and Cardiovascular Disease, University of Sheffield, Sheffield S10 2TN, UK; a.j.swift@sheffield.ac.uk (A.J.S.); s.alabed@sheffield.ac.uk (S.A.); 6Department of Clinical Radiology, Sheffield Teaching Hospitals NHS Foundation Trust, Sheffield S10 2JF, UK; 7Department of Radiology, Leiden University Medical Center, 2333 ZA Leiden, The Netherlands; r.j.van_der_geest@lumc.nl; 8Duke-NUS Medical School, National University of Singapore, 8 College Road, Singapore 169857, Singapore

**Keywords:** AI, aorta, flow displacement, validation

## Abstract

(1) *Background and Objectives*: Flow assessment using cardiovascular magnetic resonance (CMR) provides important implications in determining physiologic parameters and clinically important markers. However, post-processing of CMR images remains labor- and time-intensive. This study aims to assess the validity and repeatability of fully automated segmentation of phase contrast velocity-encoded aortic root plane. (2) *Materials and Methods:* Aortic root images from 125 patients are segmented by artificial intelligence (AI), developed using convolutional neural networks and trained with a multicentre cohort of 160 subjects. Derived simple flow indices (forward and backward flow, systolic flow and velocity) and complex indices (aortic maximum area, systolic flow reversal ratio, flow displacement, and its angle change) were compared with those derived from manual contours. (3) *Results*: AI-derived simple flow indices yielded excellent repeatability compared to human segmentation (*p* < 0.001), with an insignificant level of bias. Complex flow indices feature good to excellent repeatability (*p* < 0.001), with insignificant levels of bias except flow displacement angle change and systolic retrograde flow yielding significant levels of bias (*p* < 0.001 and *p* < 0.05, respectively). (4) *Conclusions*: Automated flow quantification using aortic root images is comparable to human segmentation and has good to excellent repeatability. However, flow helicity and systolic retrograde flow are associated with a significant level of bias. Overall, all parameters show clinical repeatability.

## 1. Introduction

Cardiovascular magnetic resonance (CMR) is the gold standard for non-invasive cardiac assessment. The most widespread method for measuring flow with CMR is through-plane phase-contrast velocity mapping. Two-dimensional (2D) phase-contrast imaging relies on the encoding of electrocardiogram (ECG) gated, single-direction (through-plane) velocity to measure cardiovascular parameters such as peak velocities, mean velocities, forward flow, regurgitant flow (or regurgitant fraction), stroke volume, and shunt volumes. In addition, aortic dimensions, mainly the aortic area, can also be assessed at that level using the magnitude images [1].

The pulsatile flow within the aorta exhibits a complex, multidirectional pattern due to the intricate structure of the aortic valve and the curved, tapering, and branching nature of the ascending aorta. The flow in the ascending aorta is also shaped by the compliance and elasticity of the aortic wall, as well as the geometry and function of the aortic valve. Additionally, the mechanical force generated by left ventricular contraction during systole, which creates a pressure gradient across the valve to propel blood forward, significantly influences the aortic flow profile and its potential hemodynamic implications.

During systole, aortic flow is primarily laminar, with a potential minor component of helical (or spiral) flow. The tricuspid aortic valve, located at the aortic root, facilitates a centralized flow with an almost uniform velocity profile entering the ascending aorta. The unique pattern of LV contraction, characterized by an anticlockwise rotation of the apical segments and a clockwise rotation of the basal segments, contributes to the helicity within the laminar flow.

In healthy individuals, the LV outflow tends to produce flow patterns that are skewed towards the inner curvature of the aorta, leading to a mild right-handed helical flow in the ascending aorta and aortic arch. Research indicates that in pathological conditions affecting the aorta, the presence of helical and vortical flows can become significantly more pronounced.

Till recently, four-dimensional flow CMR is the main imaging modality to make visual and quantitative assessments of complex aortic flow patterns. However, more recently, 2D phase contrast flow imaging has been used to assess complex aortic flow—mainly informing flow eccentricity and flow vorticity. Two such flow indices include aortic flow displacement and aortic flow reversal ratio. Aortic flow displacement refers to the eccentricity of blood flow within the ascending aorta. Flow displacement could potentially play an important clinical role in assessing the risk of aortic pathology, particularly in patients with aortic valve abnormalities and aortic dilatation. Studies have shown that patients with higher flow displacement values are more likely to experience rapid aortic growth. For instance, Burris et al. demonstrated that systolic flow displacement correlates strongly with future ascending aortic growth in patients with bicuspid aortic valves (BAV). Sigovan et al. found that the highest flow displacement values were observed in patients with stenotic tricuspid aortic valves and aortic dilation, indicating a more severe alteration in flow dynamics [2]. Increased flow displacement is associated with altered wall shear stress, which can contribute to aortic wall remodeling and aneurysm formation. Kauhanen et al. reported that patients with dilated ascending aortas had significantly higher flow displacement and circumferential wall shear stress compared to those without dilatation [3]. Flow displacement can serve as a predictive marker for aortic disease progression. The American Association for Thoracic Surgery guidelines highlight the importance of flow displacement in risk stratification for patients with BAV-related aortopathy [4]. Aortic flow reversal ratio is associated with flow vorticity in the ascending aorta. Recent literature has shown how ascending aortic flow reversal ratio during systole is pathological and could contribute to reduced aortic function. Flow reversal in the ascending aorta during systole can be caused by several factors, including aortic dilatation and adverse remodeling, aortic valve morphology and opening direction, and possibly due to left ventricular impairment resulting in early pressure equilibrium in the ascending aorta during systolic phases [5].

However, these flow indices are not routinely done. Moreover, their assessment is not easy and can be time-consuming, limiting their clinical translation and broader adoption. Automated solutions to compute all aortic flow indices, including aortic flow, flow displacement, and flow reversal ratio, do not currently exist.

The application of artificial intelligence (AI) methods in cardiovascular imaging has increased in recent years, with a particular interest in automated segmentation of short-axis cine images [6,7,8,9,10]. A few studies proposed automated aortic quantification using 2D phase contrast aortic images, demonstrating comparable accuracy to manual segmentation [11,12]. Moreover, a semi-automated systolic flow reversal ratio study using four-dimensional flow images was proposed to assess its association with aortic dilation and aortic valve stenosis, with excellent agreement between automated and manual metrics [13]. However, systolic flow displacement and its angle change assessment were not featured in any of the AI models. Assessment of these advanced metrics using AI will allow a better understanding of aortic physiology in routine clinical practice and its broader clinical translation.

We hypothesize we can develop an AI deep learning algorithm to segment aortic root and quantify normal aortic flow indices and advanced complex aortic flow indices that inform helicity, vorticity, and eccentricity of flow.

In this study, we aim to evaluate the validity and repeatability of fully automated aortic root quantification on flow, velocity, flow displacement, and its angle change using the in-house-built research software package MASS.

## 2. Materials and Methods

### 2.1. Study Cohort

We identified patients from the PREFER-CMR registry (ClinicalTrials.gov: NCT05114785) in Norfolk and Norwich University Hospitals. This study included 125 patients from Norwich University Hospital in Norwich, UK. These patients were recruited randomly from the registry.

Eligible participants for this CMR study were adults aged 18 years or older with diagnosed or suspected cardiovascular conditions. They were capable of understanding the study procedures and providing informed consent, as well as willing and able to comply with all study requirements. Additionally, participants had no contraindications to magnetic resonance imaging (MRI), such as non-MRI-compatible implants, and had stable clinical conditions without any recent major cardiovascular events.

Participants were excluded if they were pregnant, had severe renal impairment (eGFR below 30 mL/min/1.73 m^2^), or possessed any non-MRI-compatible implants or devices. Individuals who had experienced major acute cardiovascular events within the past 7 days, had uncontrolled or severe arrhythmias, or had a known allergy to gadolinium-based contrast agents (if applicable) were also excluded. Additionally, those unable to provide informed consent due to cognitive impairment or other reasons, as well as those with any serious medical conditions or comorbidities that could have interfered with the study or posed significant risks, were not eligible for participation.

### 2.2. Patient and Public Involvement

Patients hold invaluable knowledge about their personal experiences with a particular condition. They understand the daily challenges, treatment burdens, and unmet needs in a way no textbook can replicate. Involving them in study design ensures that research questions directly address issues of real concern to patients, not just researchers. Patient and public involvement (PPI) was integrated from the outset of the project via the Norfolk and Suffolk Primary and Community Care Research Office (https://nspccro.nihr.ac.uk/working-with-us/public-patient-and-carer-voice-in-research [accessed on 1 February 2024]). The PPI panel contributed to the refinement of the study protocol and the creation of patient information leaflets, ensuring they were patient-centric and encouraging open-access publication. Focus groups, interviews, and workshops provide platforms for open dialogue, allowing patients to voice their priorities, share insights on study feasibility, and suggest practical improvements to research protocols.

### 2.3. Ethics Approval and Consent to Participate

The ethical framework for this study was established according to the 2013 Declaration of Helsinki, ensuring the protection of participant rights and well-being throughout the research process. The National Research Ethics Service reviewed and approved the study’s data collection and management procedures (approval number 21/NE/0149). Additionally, all participants voluntarily provided informed consent through a streamlined opt-out process, adhering to ethical research practices as outlined in the relevant literature [14,15]. This commitment to ethical research safeguards the rights and interests of study participants while enabling valuable scientific inquiry.

### 2.4. Cardiac Magnetic Resonance and Protocol

CMR imaging was performed using a 1.5 Tesla (T) system (Magnetom Avanto, Siemens Healthcare, Erlangen, Germany), which was equipped with an eighteen-channel cardiac phase-array receiver. All patients were examined in the supine position, entering the scanner headfirst, and were monitored using a respiratory sensor along with ECG gating to synchronize the imaging with their cardiac cycle. The CMR protocol consisted of baseline surveys, cine imaging sequences, and phase-contrast 2D imaging of the aortic root.

The acquisition protocol for the two-dimensional phase-contrast images included the following parameters: an echo time of 2.44 milliseconds (ms), a repetition time of 4.53 milliseconds, and a field of view (FOV) of 380 × 296 millimeters (mm) with a coverage of 77.9%. The image resolution was set to 208 × 186 pixels, corresponding to 90% of the FOV, with a spatial resolution of 1.8 × 1.8 × 8 cubic millimeters (mm^3^). The velocity encoding parameter was set to 200 centimeters per second (cm/s), and a total of 20 cardiac time frames were acquired, with each slice having a thickness of 8 millimeters.

If flow imaging artifacts were present, they were addressed through a combination of adjusting the imaging parameters, optimizing the patient’s breath-holding technique, and, when necessary, repeating the scan to minimize the effects of motion or flow-related distortions.

### 2.5. Aortic Root Image Analysis

Two-dimensional flow assessment through the aortic root was performed with the in-house developed MASS research software (MASS; Version 2019-EXP, Leiden University Medical Center, Leiden, The Netherlands) (Figure 1). The aortic root AI was developed using convolutional neural networks [16] and trained with a multicenter cohort of 160 subjects, of which 91 were from Sheffield, UK, and 69 from Norwich and Norfolk University Hospital, UK.

Manual contours were drawn (Figure 1b) by two investigators, H.A. (3 years of CMR experience) and R.L. (1.5 years of CMR experience). The following flow indices are recorded to test the repeatability between manual and AI-derived contours (Figure 1d,e).

Aortic flow indices derived using manual and automated pipelines are broadly described in Table 1. The simple flow indices include AO forward and backward flows, SFF (Systolic forward flow), SRF (Systolic retrograde flow), Vs_avg_ (Average velocity during systole), and Vs_peak_ (Peak velocity during systole). Other geometrical indices include AO max area (Aortic maximum area). These indices have been established for broader clinical use for the last 10–15 years.

The complex flow indices include sFRR (Systolic flow reversal ratio). The aorta should exhibit only a slight retrograde flow in the systolic phase. The presence of any such flow can illuminate vortex formations that occur along the aorta’s longitudinal axis, particularly near the inner bend of the ascending aortic root. A rise in sFRR, a consequence of the vortex-induced retrograde flow, signifies less than ideal functionality of the aortic conduit [5].

FDs_avg_ (Flow displacement systolic average) is a flow parameter that can be assessed using 2D phase contrast or 4D flow CMR to quantify aortic flow eccentricity. It is calculated as the distance between the vessel centerline and the center of the eccentric flow and is normalized for overall vessel size. FDs_avg_ is a more reliable quantitative parameter for measuring eccentric aortic systolic flow than flow jet angle [17]. We also computed the ΔRA, which is the flow displacement rotational angle change between the end-systolic point and the point where the flow angle stabilized after peak systole. All these indices provide comprehensive information about the aortic flow.

### 2.6. Statistical Analysis

Data analyses were performed using SPSS (version 28.0, IBM, Chicago, IL, USA) and confirmed in MedCalc (MedCalc Software, Ostend, Belgium, version 20.011). Continuous variables were expressed as mean ± standard deviation (SD). Normality and lognormality testing were performed for all data using the Shapiro–Wilk test before the analysis. The Pearson correlation coefficient was used to calculate the correlation between manual and AI-derived flow indices. Bland–Altman plots were constructed to evaluate the agreement between manual and AI contours. Inter-observer correlation coefficient (ICC) estimates and associated 95% confidence intervals were calculated based on the absolute-agreement, 2-way mixed-effects model. A *p*-value < 0.05 is considered statistically significant.

## 3. Results

Table 2 presents the demographics and clinical data of the 125 study participants. The majority of the participants were male (62.4%). The average age was 56 years, with a standard deviation of 17.4 years. The participants’ average height and weight were 172 cm (±9.8 cm) and 82 kg (±17.7 kg), respectively, resulting in an average body surface area of 1.95 m^2^ (±0.21 m^2^). This study included 16 participants with diabetes mellitus, 38 with hypertension, 19 who had experienced a myocardial infarction, and 18 with atrial fibrillation. Additionally, 46 participants were smokers, and 32 had ischaemic heart disease. Manual and AI-deriving contours were possible in all subjects.

### 3.1. Correlation and Repeatability

AI-derived flow indices correlated strongly with those derived from manual contours (Table 3), with *p* < 0.001. aortic forward flow achieved a perfect correlation between AI and manual derivation (r = 0.996). Followed by aortic backward flow (r = 0.984), systolic retrograde flow (r = 0.969), systolic flow reversal ratio (r = 0.968), aortic max area (r = 0.964), peak systolic velocity (r = 0.947), systolic forward flow (r = 0.918), and average systolic velocity (r = 0.856). Flow displacement and helicity flow indices, including FDs_avg_ (r = 0.687), FDls_avg_ (r = 0.783), and ΔRA (r = 0.79), are the three least strongly correlated flow indices.

All ICC tests yielded *p* values < 0.001, in which seven flow indices presented excellent agreement between manual and AI contours (Table 3), including aortic forward flow [0.997, confidence interval (CI) 0.996–0.998], aortic backward flow (0.992, CI 0.989–0.994), systolic retrograde flow (0.984, CI 0.977–0.989), systolic flow reversal ratio (0.984, 0.976–0.988), aortic max area (0.982, CI 0.974–0.987), peak systolic velocity (0.973, CI 0.962–0.981), systolic forward flow (0.957, CI 0.977–0.989), and average systolic velocity (0.916, CI 0.881–0.941). Good agreements were achieved in flow displacement and helicity flow indices, ΔRA (0.882, CI 0.832–0.917), FDls_avg_ (0.814, CI 0.814–0.908), and FDs_avg_ (0.785, CI 0.694–0.849).

### 3.2. Bland–Altman Test Results

Bland–Altman plots for all recorded flow indices are presented in Figure 2. Despite achieving good to excellent repeatability in the ICC test, the comparison of ΔRA derived from manual and AI contours was associated with a statistically significant level of bias of 6.8°, followed by systolic retrograde flow, which was associated with a statistically significant level of bias of −0.3 mL (Table 4). Aside from these two flow indices, all other flow indices were associated with low levels of bias when compared manually to AI contours.

## 4. Discussion

Our study sought to investigate the validity and the repeatability of derived flow indices between manual contouring and automated AI contouring using 2D phase contrast aortic root CMR images. We have demonstrated that automated AI contour-derived flow indices strongly correlate with manual contour-derived indices. Furthermore, automated aortic root contouring presented to have good to excellent or excellent interobserver reparability compared to manual contouring with a low level of bias, though the flow displacement and its angle change assessment yielded good or excellent interobserver reparability with a relatively higher level of bias. The increased variability in advanced flow metrics is primarily due to their susceptibility to small changes in aortic area and contouring, leading to significant variations. However, the composite flow metrics, forward and backward flows, are not susceptible to the same limitation. This is the first study to use AI methods to quantify not only standard aortic flow indices but also advanced indices, which can inform about flow eccentricity, vorticity, and helicity.

There has been a rising number of studies reporting the use of AI methods for the segmentation of CMR images, and they hold promise for clinical use [7,8,9,18,19,20,21]. However, recent research showed that the majority of AI studies in cardiac imaging focus on short-axis CMR images [22]. Aortic flow assessment has important implications in determining physiologic parameters and clinically important markers associated with cardiac disease and mortality [23]. Two recent studies proposed fully AI-automated segmentation of phase contrast aortic images to derive flow and velocity indices and reported robust performance compared to manual segmentation with small differences [11,12]. Our study further evidenced that automated aortic root segmentation shows robust performance on flow displacement and helicity assessment.

Flow displacement can quantitatively represent outflow asymmetry, as evidenced by several studies to detect altered systolic flow patterns in patients with aortic valve diseases [24,25,26]. A recent study using 2D phase contrast CMR images investigated the association between aortic flow changes and exercise capacity in the context of aging without a specific cardiovascular disease [27]. Results revealed that semi-automated flow displacement and systolic flow reversal ratio had a better association with peak oxygen uptake compared to pulse wave velocity. Although flow displacement assessment shows promise in diagnostic and prognostic use, the time-intensive CMR segmentation hampers validation in a larger-scale cohort. Our study shows that automated AI contour-derived flow displacement parameters hold good repeatability to manual segmentation and provide a time-effective way to assess flow displacement in a large dataset. Moreover, our study further evaluates the reproducibility of fully automated simple and complex flow indices compared to manually derived indices.

There is a necessity for effective tools that can process and quantify relevant flow parameters and patterns. Creating a robust AI-driven post-processing pipeline can aid in incorporating sophisticated flow evaluations into regular clinical procedures, thereby broadening their use and making these techniques more accessible. Furthermore, there is an ongoing requirement for studies that validate these methods and large-scale collaborative efforts to determine the clinical usefulness and predictive power of advanced assessments of aortic flow. Studies that compare the additional benefits of metrics based on flow to traditional imaging parameters are essential to ascertaining the clinical importance of these metrics and their influence on patient care.

We acknowledge the limitations of this study. Firstly, the validation cohort of 125 subjects enrolled in this study was scanned over seven years, in which the image quality could differ from case to case and may impact the quality of the AI-derived contour. However, the repeatability of all flow indices yielded good or excellent scores. The variant image quality of CMR images also demonstrated the robust repeatability of derived parameters from the automated segmentation. Secondly, the aortic metrics from automated contours used aortic root level 2D phase contrast images. It should be noted that the repeatability of aortic flow parameters acquired from other levels could be different, such as ascending aorta and descending aorta. In addition, our cohort did not include any patients with aortic dissections; hence, our AI model is unlikely to work in that disease cohort. Furthermore, the AI model was developed using CMR scans from two centers in the UK. Although scans from both GE and Siemens scanners were employed, we should note that the AI model may not be able to incorporate different populations and pathologies. However, the MASS AI model is constantly being trained and developed using more scans from a diverse multicentre dataset to provide a robust automated post-processing solution for research. Importantly, our study did not establish the clinical value of both normal and complex flow indices. This is something that needs broader attention, and future studies will need to evaluate normal variations and disease-state aortic flow abnormalities. The methods proposed in this work can be used to phenotype large datasets, leading to further evaluation of the clinical impact of aortic flow in arthropathies. Finally, we did not include patients with significant aortic root disease—our work still warrants further investigation in patients with bicuspid aortic valve disease or co-arctation or dilation ascending aortic root.

## 5. Conclusions

This study shows that automated quantification of flow, velocity, flow reversal ratio, and flow displacement using 2D phase contrast CMR has good or excellent repeatability with a low level of bias. However, the flow displacement angle change has good repeatability with a higher level of bias. Overall, all parameters show clinical repeatability.

## Figures and Tables

**Figure 1 medicina-60-01618-f001:**
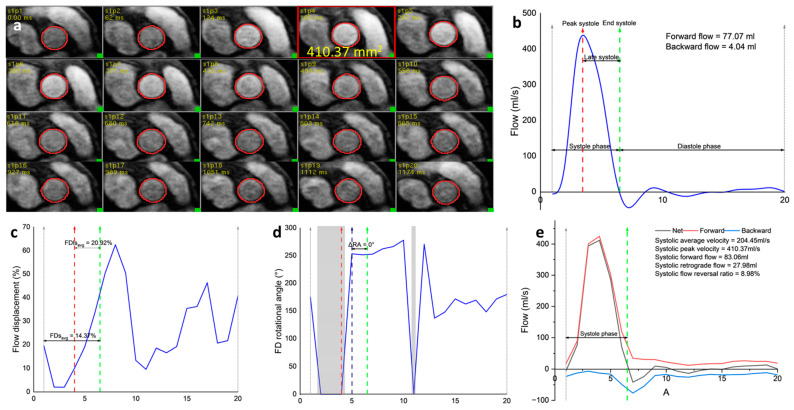
Illustration of aortic flow indices calculations. (**a**) Automated segmentation of phase contrast aortic root for the whole cardiac cycle. The red rectangle-outlined frame presents where the maximum aortic area is detected; (**b**) Aortic flow curve with illustrations of peak systole, late systole, systole, and diastole phases; (**c**) Flow displacement assessment; (**d**) Flow displacement rotational angle assessment. The gray areas denoted flow displacement ≤ 12% and were excluded in the calculations of rotational angle and rotational speed; (**e**) Flow reversal ratio assessment. The unit of the x-axis in each figure is the frame number.

**Figure 2 medicina-60-01618-f002:**
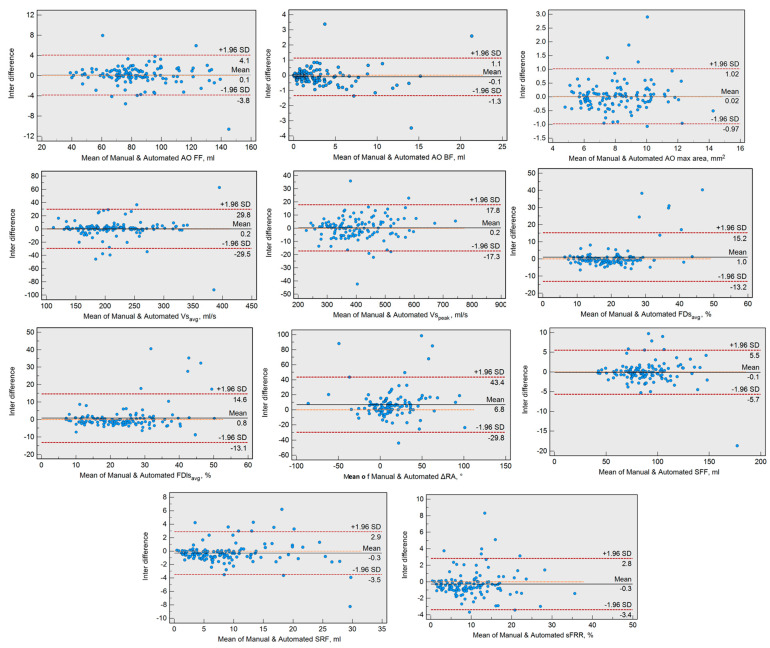
Bland–Altman plots for recorded flow indices.

**Table 1 medicina-60-01618-t001:** Recorded simple and complex flow indices.

Flow Indices	Description
Simple flow indices	
AO forward flow (mL)	Aortic forward flow
AO backward flow (mL)	Aortic backward flow
SFF (mL)	Systolic forward flow
SRF (mL)	Systolic retrograde flow
Vs_avg_ (cm/s)	Average velocity during systole
Vs_peak_ (cm/s)	Peak velocity during systole
Complex flow indices	
AO max area (mm^2^)	Aortic maximum area
sFRR (%)	Systolic flow reversal ratio
FDs_avg_ (%)	Flow displacement systolic average
FDls_avg_ (%)	Flow displacement late systolic average
ΔRA (°)	Flow displacement rotational angle change between the end-systolic point and the point the flow angle stabilized after peak systole

**Table 2 medicina-60-01618-t002:** Study participant demographics and clinical data.

	*n* = 125
Male, n (%)	78 (62.4%)
Age (years)	56 ± 17.4
Height (cm)	172 ± 9.8
Weight (kg)	82 ± 17.7
Body surface area (m^2^)	1.95 ± 0.21
Diabetes mellitus (n)	16
Hypertension (n)	38
Myocardial infarction (n)	19
Atrial fibrillation (n)	18
Smoker (n)	46
Ischaemic heart disease (n)	32

**Table 3 medicina-60-01618-t003:** Descriptive statistics of recorded flow indices derived by AI and manual contours, Pearson correlation results, and inter-observer correlation coefficient with confidence interval.

	Manual	AI	Correlation	ICC (CI)	*p*
Simple flow indices					
AO forward flow (mL)	83.83 ± 22.42	83.72 ± 22.64	0.996	0.997 (0.996–0.998)	<0.001
AO backward flow (mL)	2.98 ± 3.50	3.08 ± 3.55	0.984	0.992 (0.989–0.994)	<0.001
SFF (mL)	88.80 ± 22.94	88.03 ± 24.14	0.918	0.957 (0.940–0.970)	<0.001
SRF (mL)	8.82 ± 6.45	9.12 ± 6.58	0.969	0.984 (0.977–0.989)	<0.001
Vs_avg_ (cm/s)	212.81 ± 58.77	215.62 ± 55.84	0.856	0.916 (0.881–0.941)	<0.001
Vs_peak_ (cm/s)	403.74 ± 98.39	405.90 ± 93.13	0.947	0.973 (0.962–0.981)	<0.001
Complex flow indices					
AO max area (mm^2^)	8.11 ± 1.90	8.10 ± 1.86	0.964	0.982 (0.974–0.987)	<0.001
sFRR (%)	9.84 ± 6.30	10.12 ± 6.26	0.968	0.984 (0.976–0.988)	<0.001
FD_Savg_ (%)	20.26 ± 9.96	19.23 ± 6.95	0.687	0.785 (0.694–0.849)	<0.001
FDl_Savg_ (%)	23.97 ± 11.37	23.21 ± 9.33	0.783	0.869 (0.814–0.908)	<0.001
ΔRA (°)	15.64 ± 29.45	8.81 ± 28.08	0.790	0.882 (0.832–0.917)	<0.001

Data were represented as mean ± SD. The correlation and ICC test for all indices yielded *p* < 0.001. AO, aorta; FDs_avg_, average flow displacement during systole; FDls_avg_, average flow displacement during late systole; ΔRA, the flow displacement rotational angle change between the end-systolic point and the point where the flow angle stabilized after peak systole; Vs_avg_, average velocity during systole; Vs_peak_, peak velocity during systole; SFF, systolic forward flow; SRF, systolic retrograde flow; sFRR, systolic reversal ratio.

**Table 4 medicina-60-01618-t004:** Bland–Altman test for recorded flow indices with lower and upper limit (CI), bias, and *p*-value.

	Lower Limit (CI)	Upper Limit (CI)	Bias
Simple flow indices			
AO forward flow (mL)	−3.85 (−4.46 to −3.23)	4.07 (3.46 to 4.68)	0.11
AO backward flow (mL)	−1.34 (−1.53 to −1.15)	1.14 (0.94 to 1.33)	−0.10
SFF (mL)	−5.68 (−6.54 to −4.81)	5.52 (4.66 to 6.39)	−0.08
SRF (mL)	−3.48 (−3.98 to −2.99)	2.88 (2.39 to 3.38)	−0.30
Vs_avg_ (cm/s)	−29.48 (−34.07 to −24.89)	29.80 (25.21 to 34.39)	0.16
Vs_peak_ (cm/s)	−17.27 (−19.98 to −14.56)	17.76 (15.05 to 20.47)	0.25
Complex flow indices			
AO max area (mm^2^)	−0.97 (−1.13 to −0.82)	1.02 (0.87 to 1.17)	0.02
sFRR (%)	−3.38 (−3.86 to −2.90)	2.82 (2.35 to 3.31)	−0.28
FDs_avg_ (%)	−13.16 (−15.36 to −10.97)	15.21 (13.02 to 17.41)	1.00
FDls_avg_ (%)	−13.12 (−15.27 to −10.97)	14.65 (12.50 to 16.79)	0.76
ΔRA (°)	−29.78 (−35.44 to −24.11)	43.44 (37.78 to 49.11)	6.80

AO, aorta; CI, confidence interval; FDs_avg_, average flow displacement during systole; FDls_avg_, average flow displacement during late systole; ΔRA, the flow displacement rotational angle change between the end-systolic point and the point where the flow angle stabilized after peak systole; Vs_avg_, average velocity during systole; Vs_peak_, peak velocity during systole; SFF, systolic forward flow; SRF, systolic retrograde flow; sFRR, systolic reversal ratio.

## Data Availability

Underlying data: access to the raw images of patients is not permitted since specialized post-processing imaging-based solutions can identify the study patients in the future.

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
