# Peer review of "Automated Quantification of Simple and Complex Aortic Flow Using 2D Phase Contrast MRI"

_medicina, 2024, doi:10.3390/medicina60101618_

Round 1

Reviewer 1 Report

Comments and Suggestions for Authors

Thanks to the authors.

the authors designed a software to estimate the aortic flow rate automatically and be applied on the image analysis of the aortic root.

the aim of the work is clear.

however I need the authors to highlight the merit of the study and the addition added to the literature considering the results of this research.

Author Response

Comments 1: However I need the authors to highlight the merit of the study and the addition added to the literature considering the results of this research.

Response 1: We thank the reviewer for this insightful comment. We have added a few sentences in the discussion part paragraph 1 as the review has suggested, highlighting the merits of the study.

Reviewer 2 Report

Comments and Suggestions for Authors

The paper titled "Automated Quantification of Simple and Complex Aortic Flow Using 2D Phase Contrast MRI"explores the use of artificial intelligence (AI) to automate the segmentation of aortic root images obtained from 2D phase contrast cardiovascular magnetic resonance (CMR). The study involved 125 patients and aimed to compare AI-derived flow indices with those obtained through manual contouring. The findings indicate that AI-derived simple flow indices, such as forward and backward flow, show excellent repeatability and minimal bias compared to manual segmentation. Complex flow indices, including aortic maximum area and systolic flow reversal ratio, also demonstrate good to excellent repeatability, although parameters like flow displacement angle change and systolic retrograde flow exhibited significant bias. Overall, the study concludes that AI-based automated quantification is comparable to manual methods, with most parameters showing good to excellent repeatability, but further improvements are needed for specific indices with observed biases.

Some suggestions 

Investigate and refine the AI model for indices where significant bias was observed, particularly in flow displacement angle change and systolic retrograde flow.

Please discuss the following 

A more diverse dataset can improve the generalizability of the AI model, allowing it to perform better across different populations and pathologies

Focused improvements on these indices could enhance the clinical applicability and reliability of AI-based assessments, ensuring that they are robust enough for routine clinical use.

Understanding the real-world clinical utility and improving the integration process can promote the adoption of AI technologies in clinical practice, offering a more streamlined and efficient workflow.

Compare the results of 2D phase contrast CMR with more advanced 4D flow CMR techniques to highlight the strengths and limitations of the proposed method.

Author Response

Comments 1: Investigate and refine the AI model for indices where significant bias was observed, particularly in flow displacement angle change and systolic retrograde flow.

Response 1: We thank the reviewer for this insightful comment. The reviewer has rightly pointed out a crucial limitation of advanced flow methods, a factor that could significantly impact our field of medical research. The increased variability in advanced flow metrics is primarily due to their susceptibility to small changes in aortic area and contouring, leading to significant variations. However, the composite flow metrics, forward flow and backward flow, are not susceptible to the same limitation. We can correct this by further refining the AI model, but it is an inherent problem. Hence, we have added this in the discussion about how these advanced flow methods could be susceptible to small differences in segmentation and should always be considered when factoring them into clinical workflow. We hope that reassures the insightful reviewer.

Comments 2: A more diverse dataset can improve the generalizability of the AI model, allowing it to perform better across different populations and pathologies

Response 2: We thank the reviewer for this insightful comment and agree with it. The MASS AI model is constantly being trained and developed using more scans from a diverse multicenter dataset to provide a robust automated post-processing solution. We have added additional discussion in the limitation part to incorporate this comment on page 8, fourth paragraph.

Comments 3: Focused improvements on these indices could enhance the clinical applicability and reliability of AI-based assessments, ensuring that they are robust enough for routine clinical use.

Response 3: We completely agree that some of these indices might need further development. However, this is the initial iterative development process we have carried out for clinical applicability and reliability. In these scenes, irrespective of some degree of coefficient of variability, we still believe these indices would offer mechanistic and prognostic insight into several cardiovascular disorders and associated aortic flows.

Comments 4: Understanding the real-world clinical utility and improving the integration process can promote the adoption of AI technologies in clinical practice, offering a more streamlined and efficient workflow.

Response 4: We thank the reviewer for this insightful comment and agree with it. We discussed this in the discussion part on page 8, third paragraph: ‘There is an ongoing requirement for studies that validate these methods and large-scale collaborative efforts to determine the clinical usefulness and predictive power of advanced assessments of aortic flow. Studies that compare the additional benefits of metrics based on flow to traditional imaging parameters are essential to ascertain the clinical importance of these metrics and their influence on patient care.’

Comments 5: Compare the results of 2D phase contrast CMR with more advanced 4D flow CMR techniques to highlight the strengths and limitations of the proposed method.

Response 5: We thank the reviewer for this insightful comment and completely agree with this. However, this is outside the scope of this manuscript. We are in the process of doing another research paper comparing the results of 2D PC with 4D CMR, where we validate this against 34 metrics and is currently in press.

Round 2

Reviewer 2 Report

Comments and Suggestions for Authors

Happy with the revised version